# Molecular Mechanism of a FRET Biosensor for the Cardiac Ryanodine Receptor Pathologically Leaky State

**DOI:** 10.3390/ijms241612547

**Published:** 2023-08-08

**Authors:** Bengt Svensson, Florentin R. Nitu, Robyn T. Rebbeck, Lindsey M. McGurran, Tetsuro Oda, David D. Thomas, Donald M. Bers, Razvan L. Cornea

**Affiliations:** 1Department of Biochemistry, Molecular Biology, and Biophysics, University of Minnesota, Minneapolis, MN 55455, USA; svens005@umn.edu (B.S.); rrebbeck@umn.edu (R.T.R.); lindsey3121@gmail.com (L.M.M.);; 2Department of Pharmacology, University of California, Davis, CA 95616, USA

**Keywords:** sarcoplasmic reticulum, calcium release channel, FKBP12.6, calmodulin, FRET, fluorescence lifetime

## Abstract

Ca^2+^ leak from cardiomyocyte sarcoplasmic reticulum (SR) via hyperactive resting cardiac ryanodine receptor channels (RyR2) is pro-arrhythmic. An exogenous peptide (DPc10) binding promotes leaky RyR2 in cardiomyocytes and reports on that endogenous state. Conversely, calmodulin (CaM) binding inhibits RyR2 leak and low CaM affinity is diagnostic of leaky RyR2. These observations have led to designing a FRET biosensor for drug discovery targeting RyR2. We used FRET to clarify the molecular mechanism driving the DPc10-CaM interdependence when binding RyR2 in SR vesicles. We used donor-FKBP12.6 (D-FKBP) to resolve RyR2 binding of acceptor-CaM (A-CaM). In low nanomolar Ca^2+^, DPc10 decreased both FRET_max_ (under saturating [A-CaM]) and the CaM/RyR2 binding affinity. In micromolar Ca^2+^, DPc10 decreased FRET_max_ without affecting CaM/RyR2 binding affinity. This correlates with the analysis of fluorescence-lifetime-detected FRET, indicating that DPc10 lowers occupancy of the RyR2 CaM-binding sites in nanomolar (not micromolar) Ca^2+^ and lengthens D-FKBP/A-CaM distances independent of [Ca^2+^]. To observe DPc10/RyR2 binding, we used acceptor-DPc10 (A-DPc10). CaM weakens A-DPc10/RyR2 binding, with this effect being larger in micromolar versus nanomolar Ca^2+^. Moreover, A-DPc10/RyR2 binding is cooperative in a CaM- and FKBP-dependent manner, suggesting that both endogenous modulators promote concerted structural changes between RyR2 protomers for channel regulation. Aided by the analysis of cryo-EM structures, these insights inform further development of the DPc10-CaM paradigm for therapeutic discovery targeting RyR2.

## 1. Introduction

In failing hearts, increased diastolic Ca^2+^ leak from the sarcoplasmic reticulum (SR), via a dysfunctional hyperactive resting state of the type 2 ryanodine receptor (RyR2) channel, has been identified as a key contributor to elevated intracellular [Ca^2+^] that can trigger potentially fatal arrhythmias [1]. RyR channels are homotetrameric assemblies of 565-kDa protomers, with approximately 80% of their mass in the cytosolic headpiece domain that contains binding sites for much smaller, soluble modulators such as calmodulin (CaM) and the FK-506 binding proteins (FKBP 12.0 and 12.6) [2].

CaM is a 148-residue cytosolic protein that binds Ca^2+^ at four EF-hand sites and undergoes conformational changes that can transduce intracellular [Ca^2+^] signals into functional responses of CaM’s binding partners. Both Ca^2+^-loaded and Ca^2+^-free (apo-)CaM bind to RyRs with high affinity and a stoichiometry of 4CaM/RyR channel to partially inhibit the RyR2 Ca^2+^ channel activity [3]. The importance of CaM-mediated inhibition of RyR2 for normal heart function, is highlighted by reduced CaM/RyR2 binding affinity in heart failure (HF) and other pathophysiological states that are accompanied by a more oxidative intracellular environment [4,5,6,7,8]. Furthermore, restoring CaM binding in HF models has been shown to restore normal intracellular Ca^2+^ homeostasis [4,7,9].

The 108-residue FKBP12.6 (FKBP) binds tightly to the RyR2 channels and is proposed to promote their closed state and cooperativity between the protomers of the RyR2 complex [10,11,12]. Conversely, the absence of FKBP is associated with poorly resolved RyR2 structures [13], and might lead to an elevated Ca^2+^ leak through a destabilized channel. However, the extent and details of the correlation between FKBP-RyR2 binding and Ca^2+^ leak through RyR2 remain controversial in the myocyte environment [2,14].

DPc10 is a 36-residue peptide, derived from the RyR2 2460–2495 segment in the central sequence region, that preferentially binds to pathologically leaky RyR2 channels in permeabilized ventricular myocytes [15,16,17]. Based on this property, an acceptor-labeled derivative of the peptide (A-DPc10) has been used in a biosensor system for therapeutic discovery [18]. Mechanistically, DPc10 has been postulated to act by binding near the RyR2 N-terminus, to promote a destabilized critical contact between the N-terminal and central hot-spot regions [19]. Functionally, binding A-DPc10 promotes the Ca^2+^ leak through RyR2, as evidenced by higher Ca^2+^-spark activity and reduced CaM binding to RyR2 in permeabilized rat cardiomyocytes [20].

Anti-arrhythmogenic agents, such as CaM and dantrolene, which reduce pathological SR Ca^2+^ leak (lower Ca^2+^-spark activity), have been shown to also reduce DPc10 binding affinity for RyR2 [4,8,21] in cardiomyocytes. Therefore, RyR2 binding sites for DPc10 and CaM are thought to be tightly coupled via a negative allosteric mechanism [20], given that DPc10 inhibits RyR2-CaM binding, and vice-versa [7]. However, structural details of the crosstalk between DPc10- and CaM-binding to RyR2 are limited. Such knowledge would be valuable for the further development of A-DPc10 and related peptides as tools for therapeutic discovery [18] targeting RyR dysfunction in clinical indications, such as cardiac and skeletal myopathies or neurodegeneration (Alzheimer’s, Parkinson’s, or Huntington’s diseases) [22].

Here we used a previously developed FRET molecular system to observe the binding of acceptor-labeled ligands near donor-labeled FKBP (D-FKBP) within the RyR2 channel complex [5,8,20,23,24,25,26]. We used FRET from D-FKBPs to acceptors on either CaM or DPc10 (A-CaM or A-DPC10) to resolve structural interactions between CaM, FKBP, and DPc10 binding to intact RyR2 in cardiac SR vesicles, at nanomolar and micromolar levels [Ca^2+^].

## 2. Results

### 2.1. Effect of DPc10 on CaM/RyR2 Binding

To specifically observe CaM binding to RyR2 in SR membranes, we measured FRET to A-CaM from D-FKBP that was pre-bound to RyR2 [27]. Donor and acceptor labels were covalently attached at residue 49 of FKBP and 34 of CaM, respectively. To determine whether DPc10 interferes with CaM binding to RyR2, we measured FRET as a function of [A-CaM] in the presence and absence of 50 μM unlabeled DPc10 (Figure 1A). At this subsaturating concentration (50–100 μM), the DPc10 peptide has been typically used to induce the leaky RyR2 state [15,17,19,28]. FRET dependence on [A-CaM] in control (no DPc10) saturation curves shows high-affinity A-CaM binding to RyR2 under both 30 nM and 300 μM Ca^2+^ conditions (Figure 1A), with a ~2.7-fold higher affinity at high Ca^2+^ (Figure 1B). In 30 nM Ca^2+^, the DPc10 peptide (50 µM) decreased the apparent A-CaM/RyR2 binding affinity by 4-fold (Figure 1B), while decreasing E_max_, the FRET efficiency, by saturating A-CaM by ~37%. In 300 µM Ca^2+^ (Figure 1A), DPc10 had no significant effect on the A-CaM/RyR2 affinity, K_d_ (Figure 1B), but decreased E_max_ by ~17%. Thus, DPc10 strongly reduces the CaM/RyR2 binding affinity at low, 30 nM (resting) Ca^2+^, but not in high, 300 μM (peak-systolic) Ca^2+^. The changes in E_max_ suggest that DPc10 causes a structural shift in the A-CaM/RyR2 complex at both [Ca^2+^]. The striking distinction between the DPc10 effects on the CaM/RyR2 binding (Figure 1A) at resting vs. peak-systolic Ca^2+^ further suggests that the interaction between CaM and DPc10 binding sites involves different structural pathways under diastolic and systolic calcium concentrations, which is consistent with the shift in the CaM binding position within RyR2 that has been observed via cryo-EM [29].

### 2.2. Fluorescence Lifetime (FLT)-Detected FRET Resolves DPc10 Effects on CaM/RyR2 Structure

The decrease in E_max_ observed after incubation with DPc10 (Figure 1) may be explained by a reduction in the occupancy of RyR2 CaM-binding sites, and/or by a change in the A-CaM conformation that affects donor–acceptor distance relationships. To determine the structural basis of the DPc10 effect on E_max_, we carried out FLT-FRET measurements with a saturating concentration of A-CaM (0.8 μM), at nanomolar and micromolar levels of Ca^2+^, in the presence and absence of DPc10 (Figure 2). Under these conditions, we probed for differences in the level of CaM/RyR2 binding and structure of CaM relative to FKBP12.6 bound to RyR2 in SR membranes. Using multi-exponential analysis of these FLT-FRET data (examples of donor fluorescence decays examples in Figure 2A), we identified a one-distance Gaussian distribution as the best-fit model to describe the distance relationship between D-FKBP and A-CaM (Figure 2B). Consistent with previous reports [26], we found little Ca^2+^-driven change in the distance between the FRET probes in the no-DPc10 controls (Figure 2B, dotted traces). Similarly, there was minimal Ca^2+^-driven effect on the distance in the presence of 50 μM DPc10 (Figure 2B, solid traces). However, DPc10 led to an increase in the distance between D-FKBP and A-CaM fluorophores of 5 Å in nanomolar Ca^2+^ (Figure 2B, compare blue traces) and 6 Å in micromolar Ca^2+^ (Figure 2B, compare red traces). We also observed a Ca^2+^-driven broadening of the Gaussian distribution full-width at half maximum (FWHM) values by 2-fold (Figure 2C). This is consistent with our previous report using a different donor–acceptor pair [24]. DPc10 appears to reduce FWHM (i.e., narrow the range of detected donor–acceptor distances), but this is only a trend (Figure 2C). Finally, DPc10 also increased (X_D-only_ in Figure 2D) the donor population that that is too far from an acceptor to participate in FRET (i.e., uncoupled donors). This change was significant in 30 nM Ca^2+^, and a trend in 30 μM Ca^2+^ (Figure 2D).

To summarize the Figure 2 analysis, the FLT detection of the DPc10 effect on FRET between D-FKBP and A-CaM in nanomolar Ca^2+^ (Figure 1) suggests that this involves (1) a lengthening of the D-FKBP/A-CaM distance, (2) a decrease in disorder, and (3) a decrease in the fractional population of donors participating in FRET (i.e., an increase in the unoccupied RyR2 CaM binding sites).

### 2.3. Effect of CaM on DPc10 Binding to RyR2

To resolve DPc10 binding to RyR2 in native cardiac SR membranes, we measured FRET from D-FKBP to acceptor labeled DPc10 (A-DPc10), with the non-fluorescent acceptor probe QXL520 attached at the N-terminus of the peptide. With FRET as an index of A-DPc10 binding to RyR2, the micromolar affinity (K_d_ = 10.3 ± 0.3 µM vs. 13.2 ± 1.3 µM, in 30 nM and 300 µM Ca, respectively; Figure 3) and similarly strong FRET (E_max_ = 0.57 ± 0.05 and 0.54 ± 0.03 in 30 nM and 300 μM Ca^2+^, respectively; Figure 3) undergoes little change between nanomolar and micromolar Ca^2+^. To gauge the effect of CaM on DPc10/RyR2 binding, the FRET sample was pre-incubated with 30 μM unlabeled CaM for 40 min before addition of A-DPc10. At both nanomolar and micromolar Ca^2+^, the presence of CaM did not significantly alter E_max_ (0.54 ± 0.03 and 0.57 ± 0.06, in 30 nM and 300 μM Ca^2+^, respectively; Figure 3) relative to the no-CaM controls. This suggests that the ensemble distance relationship between the D-FKBP and A-DPc10 probes was not significantly changed by the presence of CaM within the RyR2 complex. However, CaM significantly decreased the apparent affinity of DPc10/RyR2 binding at both nanomolar and micromolar Ca^2+^. In both Ca^2+^ conditions, the K_d_ shifts are reproducible (as indicated by t-test) but their amplitude is much larger in micromolar than in nanomolar Ca^2+^ (ΔK_d_ = 4.2 ± 0.4 μM vs. 26.7 ± 5.1 μM, in 30 nM and 300 μM Ca^2+^, respectively; Figure 3).

Unlike A-CaM binding detected as FRET from D-FKBP (Figure 1), A-DPc10 binding appears to be cooperative, with a Hill coefficient (n_H_) of ~2 in 30 nM and 300 µM Ca^2+^ (Figure 3). This suggests that DPc10 binding at one site within the RyR2 homotetramer facilitates binding at additional sites. In the presence of CaM, this cooperative binding significantly increases, as indicated by the greater n_H_ values in 30 nM and 300 µM Ca^2+^.

### 2.4. Effect of FKBP12.6 on DPc10/RyR2 Binding

FKBP12 and 12.6 are thought to promote cooperativity between RyRs while also inhibiting the RyR2 leak state [10,11,12]. We hypothesized that the observed A-DPc10 binding cooperativity (n_H_ ≈ 2 in Figure 3) depends on RyR2 being decorated with D-FKBP of functional potency similar to WT-FKBP12.6. To evaluate the effect of FKBP12.6/RyR2 functional interaction on A-DPc10 binding, we measured the dependence of FRET on [A-DPc10] using the loss-of-inhibition AF488-labeled N32C-FKBP12.6 in comparison with variants of AF488-labeled FKBP (fluorophore at positions 14, 49, or 85) that are functionally equivalent to WT-FKBP12.6 [27]. We have previously shown that the D-32-FKBP variant inhibits RyR1 activity ~60% less than WT-FKBP12.6 or its other AF488-labeled single-Cys variants (labeled at positions 14, 49, or 85) while its RyR1 binding affinity remained unchanged vs. WT-FKBP12.6 [27]. We measured similar A-DPc10 binding affinity (K_d_ ≈ 10 μM) using FRET from all four different AF488-labeled variants (14, 32, 49, and 85) of FKBP12.6 (Figure 4). However, cooperativity was significantly lower for the saturation curve of A-DPc10 measured via FRET from D-32-FKBP (n_H_ = 1.3 ± 0.03) vs. D-14-FKBP (n_H_ = 1.9 ± 0.07), D-49-FKBP (n_H_ = 2.0 ± 0.15), and D-85-FKBP (n_H_ = 1.9 ± 0.12) (Figure 4). In light of the functional effect of these fluorescently labeled FKBP12.6 variants on RyR [27], this result indicates that the inhibitory efficacy of FKBP12.6 toward channel activity is a major contributor to the propensity for cooperative binding of DPc10 to RyR channels.

## 3. Discussion

We used FRET donors bound to FKBP and acceptors bound to CaM or DPc10 to determine the allosteric structural interplay between these small, soluble polypeptides upon binding to the functional RyR2 tetrameric assembly, where they regulate its activity as a Ca^2+^ release channel. Previous studies in myocytes have uncovered the allosteric interaction between CaM and DPc10 in binding to RyR2 [8,20] in quiescent (diastolic Ca^2+^) adult rat and mouse myocytes. Here, we tested the effect of Ca^2+^ on the CaM-DPc10-RyR2 relationship. E_max_ results shown in Figure 1 suggest that DPc10 induces a structural change in the CaM/RyR2 complex in nanomolar and micromolar Ca^2+^. We used FLT-detected FRET to test this hypothesis and showed that, in both 30 nM and 300 μM Ca^2+^, DPc10 induces a significant increase in the distance between A-CaM and D-FKBP (Figure 2D) while also narrowing its distribution, suggesting a decrease in disorder (Figure 2C). These effects suggest that in the DPc10-induced leaky RyR2 structure, the CaM binding domain is shifted closer to the membrane, thus further from D-FKBP, as is apparent in recent structures of the CaM-less RyR2 [13,30].

### 3.1. Ca^2+^ Effect on A-CaM and A-DPc10 Binding

The allosteric interaction between A-CaM and A-DPc10 in RyR2 binding is strongly influenced by Ca^2+^, whereby the DPc10-mediated decrease in CaM/RyR2 binding affinity is robust in 30 nM Ca^2+^ but insignificant in 300 μM Ca^2+^ (Figure 1), while the CaM-mediated decrease in RyR2-DPc10 binding affinity is >6-fold larger in 300 μM than in 30 nM Ca^2+^ (Figure 3). These Ca^2+^-mediated effects could be due to Ca^2+^ binding either to RyR2 or to CaM. However, the former is unlikely, because the impact of Ca^2+^ on DPc10/RyR2 binding affinity is small (Figure 3) compared to the impact of Ca^2+^ on CaM-RyR2 binding (Figure 1). Taken together, results in Figure 1 and Figure 3 suggest that DPc10 reduces apo-CaM (but not Ca-CaM), affinity for RyR2; Ca-CaM (but not apo-CaM) reduces DPc10 affinity for RyR2. Moreover, the smaller, yet significant effect of CaM on DPc10/RyR2 binding in nanomolar Ca^2+^ (Figure 3 left) is consistent with previous reports proposing that a significant fraction of RyR-associated CaM is Ca^2+^-loaded even under normal diastolic [Ca^2+^] [31,32,33].

### 3.2. FKBP- and CaM-Dependent Cooperativity of DPc10 Binding to RyR2

Upon measuring A-DPc10 binding, we observed that the saturation-binding curves are cooperative (n_H_ > 1), and that CaM and FKBP increase cooperativity (Figure 3 and Figure 4). This is indicated by the n_H_ increase in the presence of CaM (vs. no-CaM) (Figure 3) and by the n_H_ reduction when using the loss-of-function D-FKBP variant (labeled at 32C). Thus, CaM binding (while not cooperative itself) increases the cooperativity of DPc10 binding, which may reflect stronger structural coupling between the protomers within the tetrameric RyR channel assembly. Similarly, FKBP12.6 binding itself is not cooperative [14], but may structurally alter RyRs in a manner sensed by DPc10 cooperativity. These findings suggest that CaM- and FKBP12.6-mediated mitigation of Ca^2+^ leak through RyR channels entails stabilizing inter-protomer structural coupling within the RyR2 homotetrameric complex. This is consistent with the significant structural changes observed using cryo-EM in RyR2 upon FKBP binding [30].

### 3.3. Revised DPc10 Binding Location and Effect on RyR2 Function

Based on our previous work on determining the location of DPc10 binding to RyR2 from trilateration of FRET measurements [25] and the current RyR2 cryo-EM structures [29], we propose a new mode of interaction between RyR2 and the exogenous A-DPc10 peptide. Assuming that the C-terminal region of DPc10 intercalates within the Helical Domain 1 (HD1), this places the acceptor fluorophore (which is N-terminally attached to DPc10) underneath HD1, in the cavity formed between the Central Domain on one RyR2 protomer and the SPRY2 domain of the adjacent protomer (Figure 5). At the time of our previous report on the RyR2 mapping of DPc10 binding [25], the sequence stretches corresponding to the DPc10 peptide had not been resolved in any of the available structures. When the high-resolution RyR2 structure became available [13,29], we realized that the FRET-determined distances were probably underestimated because the RyR2 sequence corresponding to DPc10 is centrally-located, which leads to two acceptors contributing to FRET with each donor. By re-evaluating our previous FRET data for trilateration of A-DPc10 in light of the new RyR2 cryo-EM structures, the acceptor probe can be mapped to the cavity near the SPRY2 domain. In the cryo-EM structure of the closed-state RyR2 with apo-CaM and FKBP bound (PDB ID: 6JI8) [29], which is the most relevant structure for our experimental conditions, the N-lobe of CaM is bound near HD1 and may stabilize that domain, thus reducing DPc10 binding. Based on this structure combined with our previously reported FRET results, we propose that the acceptor probe of the A-DPc10 biosensor construct is positioned closer to the C-lobe of CaM, which is bound deeper in the cleft, near the handle and central domains. Cryo-EM suggests that in both the open and closed state of RyR2, Ca-CaM, binds deeper in the cleft (than apo-CaM) and interacts with the central domain but not with HD1 [29]. In this posture, the N-lobe of Ca-CaM is further away from the putative binding site of DPc10. Taking all this into account, the potential mechanism by which DPc10 promotes a leaky RyR2 involves competition with the native helix-helix packing in HD1, thereby weakening the interaction between HD1 and NTD_A_ or HD1 and SPRY2 domains of the next protomer. From the cryo-EM structures it can be proposed that apo-CaM stabilizes the HD1 region, thereby reducing the DPc10 binding affinity. By its binding to the central domain, Ca-CaM causes a rotation of that domain which stabilizes the closed state inhibiting the RyR2 channel at high [Ca^2+^]. This important mechanism is a potential explanation of the stronger effect of Ca-CaM vs. apo-CaM on decreasing DPc10 binding (Figure 3 right vs. left panel) by stabilizing the closed state.

Our previous work in permeabilized adult rat ventricular myocytes [20] at 50 nM [Ca]_i_ showed that CaM dramatically suppressed DPc10 binding to RyR2 (in agreement with Figure 3 here), but that FKBP12.6 had little effect on DPc10 binding. In terms of diastolic RyR2 leak, DPc10 induced an increase in SR Ca leak (seen as Ca sparks) that was suppressed by saturating CaM, but not by FKBP12.6. Thus, while the binding of either CaM or FKBP12.6 to RyR2 may disfavor DPc10 binding to RyR2, the functional consequences of CaM on SR Ca leak appear to be much stronger than for DPc10.

## 4. Materials and Methods

### 4.1. Materials

SR vesicles were isolated by differential ultracentrifugation of ventricular tissue obtained from porcine hearts procured commercially [27]. Residual native CaM was stripped from SR by incubation with 300 nM of the M13 skeletal muscle myosin light-chain kinase CaM-binding peptide (Anaspec, Fremont, CA, USA), as described previously [35]. Resuspended SR (30 mg/mL) was flash-frozen and stored at −80 °C. Fluorescent FKBP and CaM were prepared as previously described [27]. Briefly, Alexa Fluor 488 (AF488; Life Technologies, Eugene, OR, USA) was used to label single cysteines substituted at positions 14, 32, 49, and 85 into a null-cysteine (C22A/76I) variant of human FKBP12.6. Alexa Fluor 555 (AF555, Life Technologies) was used to label a single cysteine substituted at position 34 in the wild-type mammalian CaM [23]. DPc10 peptides were synthesized at AnaSpec (Fremont, CA, USA) corresponding to the RyR2 sequence 2460-GFCPDHKAAMVLFLDRVYGIEVQDFLLHLLEVGFLP-2495 [36]. For FRET assays of DPc10/RyR2 binding, the dark acceptor QXL520 was covalently attached at the N-terminus of DPc10 (resulting in A-DPc10) at AnaSpec. Other reagents were from Sigma (St. Louis, MO, USA).

### 4.2. FRET Measurements

RyR2 in cardiac SR membranes was preloaded with D-FKBP as described previously [27]. Samples were then incubated with the indicated concentration of WT-CaM or DPc10 for 40 min at 22 °C in media containing 150 mM KCl, 5 mM GSH, 0.1 mg/mL BSA, 1 μg/mL Aprotinin/Leupeptin, 1 mM DTT, and 20 mM PIPES (pH 7.0), 1 mM EGTA, and 0.065 or 1.338 mM CaCl_2_ to give 30 nM or 300 μM Ca^2+^ (calculated by MaxChelator v1.0). The binding of A-CaM or A-DPc10 to RyR in the proximity of D-FKBP was measured following 2.5-hr incubation at 22 °C. Fluorescence spectra were acquired as described previously [27]. For the AF488/AF555 and AF488/QXL520 donor/acceptor pairs, distance calculations were made based on Förster distances of 70 and 54 Å, respectively.

### 4.3. Model-Based Fitting of Time-Resolved FRET Data

For the FLT detection of FRET, we used time-correlated single-photon counting as described previously [37]. Using global multi-exponential analysis, we performed multi-exponential fitting of the FLT-FRET data to a series of structural models, as extensively described elsewhere [37,38]. This type of FLT-FRET data analysis resolves both binding and structural information.

### 4.4. Statistics

Sample means are from ≥3 independent experiments (as indicated by n in figure legends). Data are presented as mean ± S.E. Statistical significance was evaluated using unpaired Student’s *t* test.

## 5. Conclusions

In this report, we demonstrate the structural impact of DPc10 on CaM/RyR2 binding and highlight the impact of DPc10 on RyR2 structure at the CaM binding sites. In addition, by examining the role of FKBP12.6 and CaM-mediated inhibition on the cooperative binding of DPc10 to RyR2 protomers, we provide novel information into the role of these native, soluble, cytosolic RyR2 regulators on the RyR2 leak state. In light of previous FRET and cryo-EM data, the new information provided here can translate into further development of biosensors of RyRs for therapeutic discovery, targeting a broad range of clinical indications fueled by dysfunctional Ca^2+^ homeostasis.

## Figures and Tables

**Figure 1 ijms-24-12547-f001:**
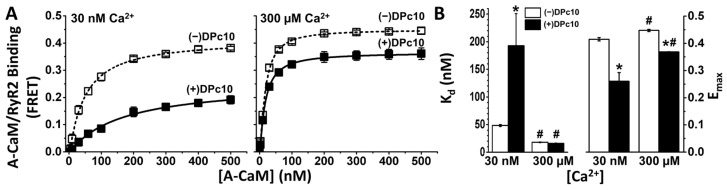
DPc10 effect on CaM/RyR2 binding. (**A**) RyR2 in SR isolated from the porcine heart was decorated with D-FKBP, then treated with (solid symbols) or without (open symbols) DPc10 (50 μM) for 60 min (25 °C). The concentration dependence of A-CaM binding to RyR2 was measured using FRET after further incubating with A-CaM at the indicated concentrations (150 min, 25 °C), in 30 nM or 300 μM Ca^2+^, as indicated. (**B**), Parameters K_d_ and E_max_ obtained from fitting the data shown in A to a single rectangular hyperbola. Data represent the mean ± SE from four paired experiments. * significantly different from no-DPc10 controls; # significantly different from the 30 nM Ca^2+^ condition, as determined by *p* ≤ 0.05 (unpaired Student’s *t*-tests).

**Figure 2 ijms-24-12547-f002:**
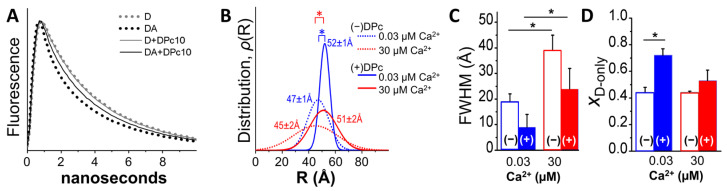
FLT-FRET detection of DPc10 structural effect on CaM/RyR2 complex. (**A**) RyR2 in cardiac SR membranes was decorated with D-FKBP, then treated with (solid line) or without (dotted line) DPc10 (50 μM), and finally incubated with saturating (0.8 μM) A-CaM. A representative FLT-FRET experiment is illustrated by fluorescence decays of D-FKBP without A-CaM (grey, D) and with (black, DA) A-CaM. (**B**) Multi-exponential analysis of panel (**A**) FLT-FRET decays resolves a one-distance Gaussian distribution of the separation between D-FKBP and A-CaM on RyR2 in 30 nM (blue) or 30 μM (red) Ca^2+^. Distributions are centered at the distances indicated in panel (**A**), and their widths (FWHMs) are shown in panel (**C**). (**D**) The effect of Ca^2+^ and DPc10 on the mole-fraction of D-FKBP not participating in FRET (*X*_D-only_). Data are shown as means ± SE; n = 4–6. * *p* ≤ 0.5 (paired Student’s *t*-test).

**Figure 3 ijms-24-12547-f003:**
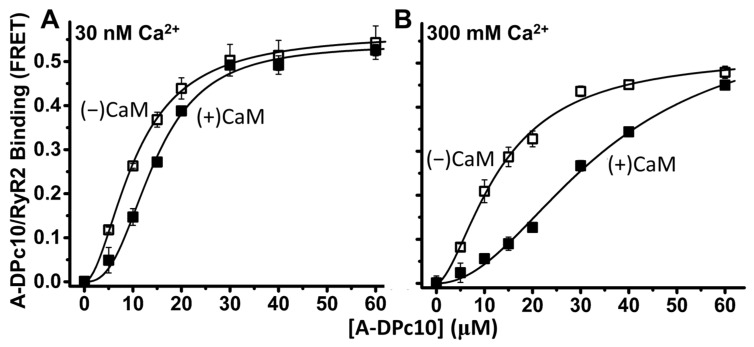
CaM influence on DPc10/RyR2 binding. RyR2 in cardiac SR membranes was decorated with D-FKBP and then pre-incubated 40 min with or without 1 μM CaM. These samples were then further incubated (40 min) with a series of [A-DPc10], as indicated, at 30 nM Ca^2+^ (**A**) or 300 μM Ca^2+^ (**B**). DPc10/RyR2 binding was measured using FRET from D-FKBP to A-DPc10. Fitting was performed using the Hill function; fitting parameters are shown in Table 1. Data are shown as mean ± SE, n = 4–6 (paired Student’s *t*-test).

**Figure 4 ijms-24-12547-f004:**
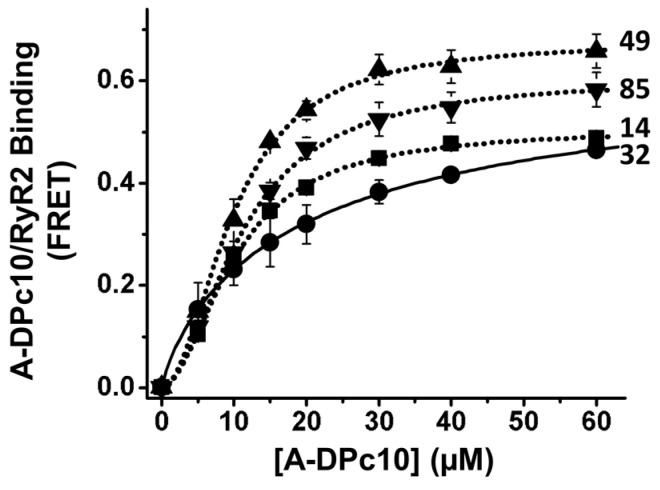
RyR2 in cardiac SR membranes were decorated with either D-14-FKBP, D-32-FKBP, D-49-FKBP, D-85-FKBP, and incubated with A-DPc10. Fitting parameters are shown in Table 2 (n = 4–6).

**Figure 5 ijms-24-12547-f005:**
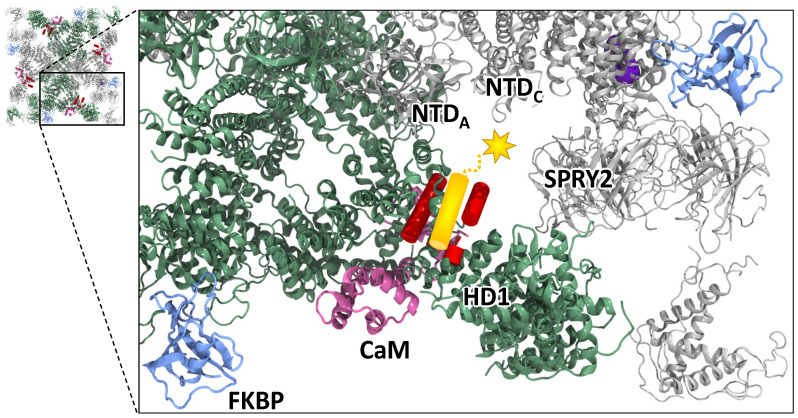
Location of the A-DPc10 biosensor within the RyR2 structure. The RyR2 cryo-EM structure with apo-CaM and FKBP bound (PDB ID: 6JI8) [29] is displayed as ribbons. Two RyR2 protomers are shown in green and the other two are shown in gray to emphasize protomer boundaries. CaM is shown in purple and FKBP is shown in blue. The red cylinders depict the helices of the RyR2 HD1 that correspond to the 2460–2495 segment, which is the same sequence as in DPc10. Depicted in yellow is A-DPc10, with a predicted C-terminal α-helix and an unstructured N-terminal stretch. The A-DPc10 helix (yellow) is modeled in contact with the red HD1 helices where DPc10 has been postulated to bind. The acceptor fluorophore is shown as a yellow star. The fluorophore is located in the cavity below HD1, NTD_A_, and NTD_C_, between the central domain of one protomer and the SPRY2 domain of the adjacent protomer. Structure illustrations were created using VMD 1.9.3 [34].

**Table 1 ijms-24-12547-t001:** Fitting parameters for curves in Figure 3 ^1^.

	30 nM Ca^2+^			300 μM Ca^2+^		
	K_D_ (µM)	E_max_	n_H_	K_D_ (µM)	E_max_	n_H_
(−) CaM	10.4 (0.3) *	0.57 (0.05)	2.0 (0.1) *	13.2 (1.3) *	0.54 (0.03)	1.8 (0.1)
(+) CaM	14.6 (0.1) *	0.54 (0.03)	2.6 (0.3) *	33.9 (3.8) *	0.57 (0.06)	2.3 (0.4)

^1^ Fitting performed using Hill function. * statistical significance (−) vs. (+) CaM.

**Table 2 ijms-24-12547-t002:** Parameters of best fit to the Figure 4 data ^1^.

D-x-FKBP ^2^	K_D_ (µM)	E_max_	n_H_
14	10.0 (0.2)	0.51 (0.01)	1.9 (0.07)
32	13.1 (0.3)	0.55 (0.04)	1.3 (0.03)
49	9.9 (0.4)	0.67 (0.02)	2.0 (0.15)
85	10.4 (0.3)	0.57 (0.05)	2.0 (0.10)

^1^ Fitting performed using Hill function (n = 4–6). ^2^
**x** indicates the labeled position within the FKBP12.6 sequence.

## Data Availability

Data are contained within the article; primary data are available from the authors upon request.

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
