# Peer review of "Molecular Mechanism of a FRET Biosensor for the Cardiac Ryanodine Receptor Pathologically Leaky State"

_ijms, 2023, doi:10.3390/ijms241612547_

Round 1
Reviewer 1 Report
A manuscript by Svensson et al. examined cross-talking between CaM and DPc10 binding to RyR2 channel using FRET measurements. The technique was originally developed by the authors’ group and has been providing a number of important biochemical insights of ryanodine receptor, especially RyR-CaM interactions. While the current data added another interesting property of RyR-CaM-DPc10 interaction, I have some concerns on the manuscript.
Authors should perform functional analysis of RyR2 in the presence of combination of CaM and DPc10 at diastolic and systolic Ca2+. For example, is RyR2 inhibition by CaM attenuated by DPc10 at diastolic Ca2+ but not at systolic Ca2+? If such experiments have been done previously, authors should discuss how the current data advances our understanding of RyR2 structure/function by comparing with previous data.
Authors should also use the mutant DPc10 with a CPVT mutation, namely an inactive DPc10, as a negative control.
Why did the authors use 30 µM CaM (100-1000 times higher than Kd) in Fig. 3?
In Fig. 5 authors used RyR2 structure with apoCaM. I think the cryo electron microscopy data set for RyR2 with CaCaM is also available. Are there any structural differences between the two which may affect the A-DPc10 binding? Authors should discuss this by comparing their FRET data in the presence of apoCaM or CaCaM.
Author Response
We have uploaded a Word file containing our responses to Reviewer 1.

Reviewer 2 Report
In this manuscript, Svenson et al. present results obtained from FRET experiments to understand molecular mechanisms and interactions of RyR2 with some endogenous modulators of this channels. These interactions are crucial to know taking into account how calcium released from RyR2 plays an important role in physiological and pathological (arrythmias) conditions in the heart. This study reveals that, depending on the calcium level, nanomolar that corresponds to resting state or micromolar range for contraction state, these modulators promote concerted structural changes for channel regulation. Understanding these mechanisms is of major interest for arrythmias and also for other diseases displaying leaky release channels or calcium homeostasis dysregulation.
Minor comments :
- Introduction : The authors should more clarify what is called a "leaky" channel and what molecular mechanisms are underneath this "leaky“ property.
- line 100: The use of "µM" or "nM" Ca2+ abbreviations directly in the sentences is unclear, maybe use "micromolar" or "nanomolar" “context” or “environment" or “levels”.
Author Response
Please find below reviewer comments in italics, and author responses in normal font.
Reviewer 2 comments:
In this manuscript, Svenson et al. present results obtained from FRET experiments to understand molecular mechanisms and interactions of RyR2 with some endogenous modulators of this channels. These interactions are crucial to know taking into account how calcium released from RyR2 plays an important role in physiological and pathological (arrythmias) conditions in the heart. This study reveals that, depending on the calcium level, nanomolar that corresponds to resting state or micromolar range for contraction state, these modulators promote concerted structural changes for channel regulation. Understanding these mechanisms is of major interest for arrythmias and also for other diseases displaying leaky release channels or calcium homeostasis dysregulation.
Minor comments :
- Introduction : The authors should more clarify what is called a "leaky" channel and what molecular mechanisms are underneath this "leaky“ property.
Response: Thank you for reminding us that the concept of a leaky channel requires careful definition. We have revised the Introduction to clarify what we mean by leaky RyR2.
- line 100: The use of "µM" or "nM" Ca2+ abbreviations directly in the sentences is unclear, maybe use "micromolar" or "nanomolar" “context” or “environment" or “levels”.
Response: We made the suggested changes (much appreciated!).
Round 2
Reviewer 1 Report
Authors improved the manuscript. It is now ready to publish.